# Influential Factors and Determination Method of Unconventional Outside Left-Turn Lanes Based on a BP Neural Network

**Yi Cao, Dandan Jiang * and Xuetong Li**

School of Transportation Engineering, Dalian Jiaotong University, Dalian 116028, China; caoyi820619@aliyun.com (Y.C.); m17866544295@163.com (X.L.)
* Correspondence: dandan20210118@163.com; Tel.: +86-159-0496-7141

**Abstract:** To reduce the delay caused by the interweaving and parallel driving of multiple left-turn vehicles and through vehicles upstream of the intersection entrance, the influencing factors and determination methods of unconventional left-turn lanes are studied in right-hand traffic (RHT) countries. For countries driving right, left-turn lanes are usually on the inside of roads. However, when there are a large number of vehicles turning left in the outer lane of the upstream section of the intersection, these vehicles will be forced to pass many consecutive parallel lanes and then enter the left-turn lane. During this process, many traffic conflicts will occur between left-turning vehicles and going-straight vehicles, which will lead to longer traffic delays. To reduce traffic conflicts and delays caused by problems mentioned before, a scheme of setting left-turn lanes abroad is proposed, and major influencing factors and judgment methods of such a scheme are also studied. With the help of traffic simulation software VISSIM, the simulation model of intersection entrance with a different number of through lanes, length of weaving section and left turn inner and outer lanes is established. By inputting different numbers of entry through vehicles and left-turning vehicles in the outer lane, the delay data under different geometric and traffic conditions are obtained for simulation analysis. With the help of MATLAB software, this paper analyzes the influence of the length of the weaving area and the number of left-turning vehicles on the delay of inside and outside left-turning lanes under the condition of a different number of straight vehicles, as well as the variation law between them. By inputting parameters such as the length of the weaving area and the number of lanes, go-straight vehicles and left-turning vehicles into the system of VISSIM, a BP neural network model is constructed and trained. When investigating the entrances of four intersections, the BP neural network model is used to analyze and calculate the traffic delay and determine the setting scheme of the inside or outside of the left-turn lane. Through experiments and further studies, a phenomenon was found: When more vehicles chose to turn left or go straight in the outside lane, the length of the weaving area will become shorter, and the delay reduction effect of the unconventional left-turn lane will more obvious. The specific location of the left-turn lane should be determined by the constructed BP neural network model through the comparative analysis of delay, and the judgment results are in good agreement with the realistic scheme.

**Keywords:** traffic engineering; route optimization; genetic algorithm; feeder bus; station transfer

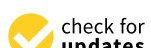



## 1. Introduction

### 1.1. Background

There are two models of road traffic all over the world: one is left-hand traffic (LHT) and the other is right-hand traffic (RHT). Most countries implement RHT, such as China, the United States, and Russia. Signalized intersections are the main places in urban traffic where traffic conflicts occur. For right-hand traffic (RHT) countries, left-turn traffic is one of the main causes of traffic conflicts. Signalized intersections are key nodes and capacity

bottlenecks in the urban road network system. Reasonable and effective organization of left-turn traffic is one important way to improve the operating status of intersections. The setting of the left-turn lane is related to the overall operating quality of the intersection, which has attracted a lot of attention for a long time. With the increase in traffic load in all directions of urban signalized intersections and the frequent occurrence of upstream lane changes in recent years, some applications of unconventional outside left-turn lane have emerged at intersections in RHT countries. On the contrary, traffic organization of right-turn vehicles at intersections and outside right-turn lanes are the main problems in LHT countries, such as the United Kingdom and Japan. Generally speaking, the essential causes of the above two problems are the same, so this paper only takes RHT countries as an example to discuss the problem of the outside left-turn lane. Relevant studies have shown that the left-turn lane has its advantages and limitations on the inside or outside of the entrance lane. In this paper, simulation experiments, artificial neural network modeling, and measured data are used to quantitatively study whether the unconventional outside left-turn lane can obtain a smaller intersection entrance lane delay than an inside left-turn lane under any conditions so as to quantitatively clarify the setting conditions of the outside left-turn lane. The research results have certain theoretical guiding significance and practical application value for the reasonable composition of left-turn lanes at intersections, the reduction in interweaving and conflict phenomena, and the reasonable organization of left-turn vehicle flow.

Relevant studies have shown that if there are more left-turn vehicles on the outside lane upstream segment of intersection entrance, the outside left-turn lane can effectively alleviate the interweaving phenomenon among these vehicles and straight-going vehicles (Li J. W. et al., 2010) [1]. However, some scholars have evaluated the safety impact of this traffic organization mode based on video traffic conflict automatic technology and found that the intersection approach with an outside left-turn lane has more conflicts than an intersection with an inside left-turn, and an improper setting will adversely affect overall operation efficiency and safety of the intersection (Guo Y. et al., 2016) [2]. Therefore, it is necessary to carry out quantitative research on the influential factors and setting conditions of the outside left-turn lane.

Although domestic and foreign scholars have conducted some research on the external placement of the left-turn lane, they mainly focus on the influence of changing only one factor—the location of the left-turn lane. The analysis of how to set the left-turn lane is usually based on the staggered behavior of the vehicle. In addition, most research on the socialization of left-turn lanes is generalized, and they ignore the special case of setting left-turn lanes. There are many research findings on using a BP neural network to solve traffic problems, but most of them are predictions of traffic flow. The gap in research on the prediction of traffic delay is still huge. In view of the above, this paper has the following two contributions to identifying external factors influencing the left-turn lane and analysis of those factors:

1. Qualitative analysis was made on the influencing factors of the setting of external left-turn lane in this paper. Factors such as traffic flow, left-turn traffic flow, weaving section length, and the number of through lanes are considered, and a thorough analysis was made about their impact on the delay of external entrance lane in the left-turn lane. To some extent, the findings in this paper fill the gap of the lack of comprehensive consideration of multiple factors in previously published literature.
2. Comparisons are made between the simulated delay value and the predicted delay value, which is the feasibility of BP neural network prediction. Such an attempt, to some degree, makes up for the prediction research of solving traffic delays based on a BP neural network.

The research results are of very important theoretical and social practical value in reducing traffic delays and traffic weaving at intersection entrances and improving the road capacity.

This paper is divided into seven parts: introduction, case, study, methodology, simulation experiment, results, discussion, and conclusion.

### 1.2. Literature Review

At present, domestic and foreign scholars have conducted relevant research on the setting of left-turn lanes. In terms of influencing factors, Ma W. et al. (2017) [3] studied the variation law of the maximum passing rate and blocking probability of the left-turn lane, adopting the approach of changing influencing factors, such as the storage capacity, cycle length, green diversion, phase sequence, and traffic flow, when the left-turn lane is blocked to find the law. Additionally, the volume and capacity of the left-turn lane are taken into account for the improvement of signalized intersections. Sando T. et al. (2009) [4] surveyed three left-turn lanes, analyzed the influence of geometric factors on the utilization rate of left-turn lanes, vehicle saturated flow, and other factors, and considered that the main factors of highly saturated flow were a downhill direction and a turning angle less than 90°. From the perspective of system design, Kikuchi S. (2021) [5] predicted the optimal length of right-turn and left-turn lanes at an intersection, and the relevant parameters affecting the length of left- or right-turn lanes were considered and analyzed. Persaud B. et al. (2010) [6] analyzed the queue length from the perspective of preventing vehicles from changing lanes due to congestion and developed a left-turn lane length calculation framework under different signal timing. Wang X. et al. (2020) [7] believed that lane width, the proportion of large vehicles, and the proportion of left-turning vehicles were important factors that affect the saturation flow of the lane. Daamen, W. (2010) [8] considered the interweaving flow rate and length important influential factors of the traffic operation at the entrance and exit. Xu Y. et al. (2018) [9] established a service level calculation model for the interweaving section and obtained the minimum weaving section length required to meet a certain service level and interweaving traffic. Han Y. et al. (2021) [10] analyzed the changes in vehicle interaction under different driving modes and then established a vehicle lane change model that can better adapt to difficult traffic conditions, which was validated through practice cases. Yao, R.H. (2009) [11] found that vehicles were more inclined to change lanes in the first half of the weaving area under crowded conditions but in the second half under free-flow conditions through investigation. Fitzpatrick Kay et al. (2014) [12] found the influential factors of geometry and traffic composition of double left-turn lanes after comparing the saturation flow rate of inside and outside left-turn lanes.

In terms of setting conditions, Wu J. et al. (2019) [13] proposed a procedure to optimize the distance between the upstream central divider opening and the main signal light using the CLL design method to allow more cars to go through the left-turn lane, thus maximizing the emission rate of left-turning vehicles and the utilization rate of the countercurrent lane. Zhou H. et al. (2010) [14] studied the saturated lane and lane distribution of inside and outside two-way left-turn lanes and found that the utilization rates of the inside and outside left-turn lanes were 46% and 54%, respectively. Zheng C. et al. (2013) [15] put forward the design method of replacing the left-turn lane and right-turn lane. They also analyzed the delay of the entrance lane in the case of a bus lane. Liu Pan et al. (2013) [16,17] quantitatively studied the impact of outside left-turn lanes on the traffic capacity and the intersection's operating efficiency with the binary Logit model. They believe that the probability of a driver choosing to drive in the outside left-turn lane increases with the increase in the traffic volume of the main line and the length of the queue in the inside left-turn lane. It decreases as the distance from the upstream right auxiliary side road to the intersection increases.

Domestic and foreign research has been conducted on the issue of outside left-turn lanes. However, most of them are limited to the impact of a single influential factor on the location of left-turn lanes, while other factors remain unchanged. They lack quantitative research on the location of left-turn lanes when multiple factors are involved. In addition, there are also special case studies on outside left-turn bus lanes, but general social lane conditions are ignored. Therefore, the delay data of inside and outside left-turn lanes are obtained under the comprehensive influence of multiple factors through the VISSIM

simulation experiment. Based on the analysis of influential factors, the BP neural network delay model is constructed to comprehensively determine the location of the left-turn lane.

## 2. Case Study

*Traffic Survey Data*

In view of the inside left-turn lane, traffic surveys were conducted at the intersection of Southwest Rd. and Nansha St. and the intersection of Southwest Rd. and Huanghe Rd. in Dalian during the peak hours of 7:00–8:00 and 17:00–18:00 on 12–13 July 2019. In view of the unconventional outside left-turn lane, traffic surveys were conducted at the intersections of Xinlong St. and Shiji Road and Xinggong St. and Shenliao Rd. in Shenyang during the peak hours of 7:00–8:00 and 17:00–18:00 on 5–6 July 2019. All the surveys were conducted manually. Four investigators were set up at each intersection and two at each of the entrances opposite the main road. During the investigation, the number of straight-going lanes on the entrance road, the length of the upstream weaving section, and the setting location of the left-turn lane were recorded firstly for both inside and outside left-turn lanes. One investigator observed and recorded the number of straight-going vehicles in the entrance lane, and the other investigator observed and recorded the number of left-turning vehicles in the outside lane. The survey distinguishes only small and large cars. Therefore, four investigators are required for each intersection, and two intersections are investigated at the same time every day. A total of 8 investigators are required. Another investigator is required for data summary and collation, as shown in Table 1. This survey was designed and made according to the research content and approved by Dalian Jiaotong University. The data used in the case study were obtained from the survey. Additionally, road conditions at the four intersections can be clearly seen in Figure 1.

**Table 1.** Survey plan.

| Location | Longitude Coordinates | Latitude Coordinates | Date | Period | Investigator (Individual) |
|---|---|---|---|---|---|
| The intersection of Southwest Rd. and Nansha St. | 121.57147 | 38.914697 | 12 July 2019 13 July 2019 | 7:00–8:00 17:00–18:00 | 8 |
| The intersection of Southwest Rd. and Huanghe Rd. | 121.575113 | 38.920412 | 12 July 2019 13 July 2019 | 7:00–8:00 17:00–18:00 | 8 |
| The intersection of Xinlong St. and Shiji Rd. | 123.46434 | 41.724777 | 5 July 2019 6 July 2019 | 7:00–8:00 17:00–18:00 | 8 |
| The intersection of Xinggong St. and Shenliao Rd. | 123.38505 | 41.788174 | 5 July 2019 6 July 2019 | 7:00–8:00 17:00–18:00 | 8 |

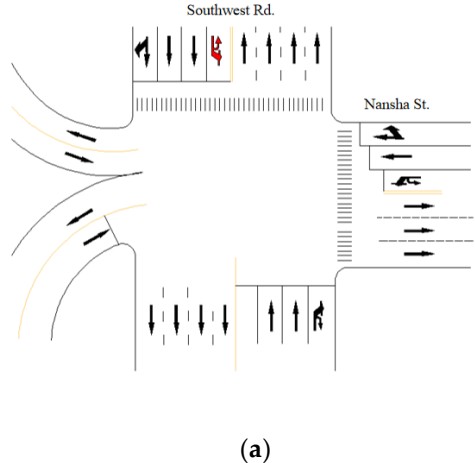

(**a**)

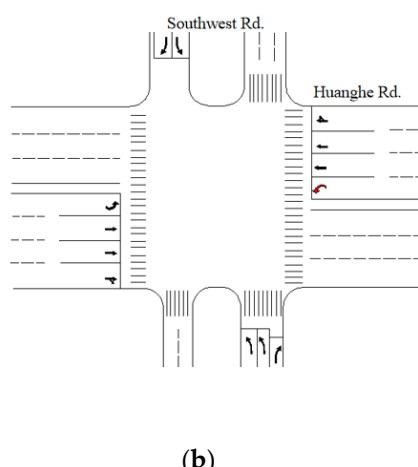

(**b**)

**Figure 1.** *Cont.*

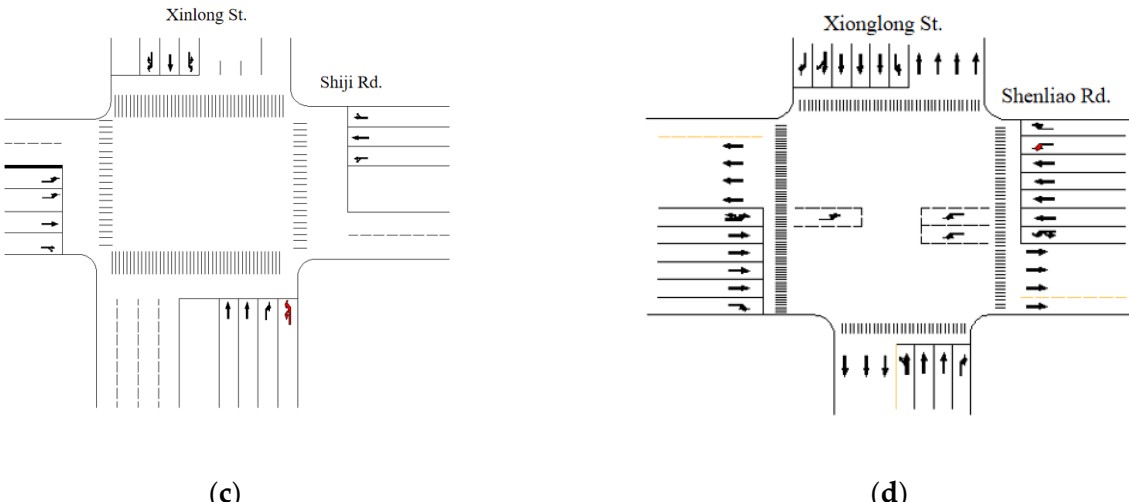

**(c)**  **(d)**

**Figure 1.** Road conditions at the four intersections. (**a**) The intersection of Southwest Rd. and Nansha St. (inside). (**b**) The intersection of Southwest Rd.and Huanghe Rd. (inside). (**c**) The intersection of Xinggong St. and Shenliao Rd. (outside). (**d**) The intersection of Xinlong St. and Shiji Rd. (outside).

## 3. Methodology

### 3.1. Analysis and of Influential Relationship

Taking the entrance lanes of two straight-going lanes as an example, four conditions are selected as the number of straight-going vehicles as follows: 600, 750, 900, and 1050 veh/h. With the help of MATLAB, the influence of the different numbers of left-turn vehicles and lengths of weaving sections on the delay of the left-turn lane inside or outside of the entrance lane is fitted and discussed. The relation curve is shown in Figure 2. The three straight-going lanes have similar influential relationships.

It can be seen from Figure 2 that when the number of straight-going vehicles is constant, the delay changing law of inside or outside left-turn lanes has certain similarities. With the increase in the number of left-turn vehicles and the decrease in the length of the weaving section, the delay will increase significantly, and the delay change of the inside lane is more obvious than that of the outside lane. The more vehicles that need to merge from outside to inside, the more left-turn vehicles there are. The shorter the length of the weaving section, the greater the probability of the above-mentioned lane changing behavior and colliding with straight-going vehicles within a shorter distance. When the number of straight-going vehicles increases, the delay of the left-turn lanes inside or outside will rise, but the inside scheme increases faster than that of the outside. Inside and outside delay surfaces have intersections, as shown through further analysis of Figure 2a–c. As shown in Table 2, The projection curve equation of the intersection line of inside and outside delay surfaces corresponding to different numbers of straight-going vehicles can be obtained by combining the inside and outside fitting surface equations, eliminating the vertical coordinates, and setting it zero. The curve equation expresses the corresponding relationship between the number of left-turn vehicles and the length of the weaving section under the critical state of equal inside and outside delay of the left-turn lane. X represents the number of left-turn vehicles, and y represents the length of the weaving section. Therefore, the advantages and disadvantages of the inside or outside left-turn lane scheme can be determined by the position relationship between the projection curve and the points formed by the number of left-turn vehicles or the length of the weaving section. For a combination of the number of left-turn vehicles and the length of the weaving section (x, y), the calculated value y′ can be obtained by substituting x into the corresponding equation in Table 2, and then y and y′ can be compared. If y > y′, the (x, y) point is located above the intersection projection, and the delay of the outside left-turn lane in the corresponding curved surface is greater than that of the inside; therefore, it is more advantageous to choose an inside left-turn lane.

On the contrary, if y < y′, the outside left-turn lane is more advantageous. The two curved surfaces in Figure 2d have no intersection. It means that the outside left-turn lane is better than the inside one when the number of straight-going vehicles is above 1050, no matter how the number of left-turn vehicles and length of weaving section are taken within their respective ranges.

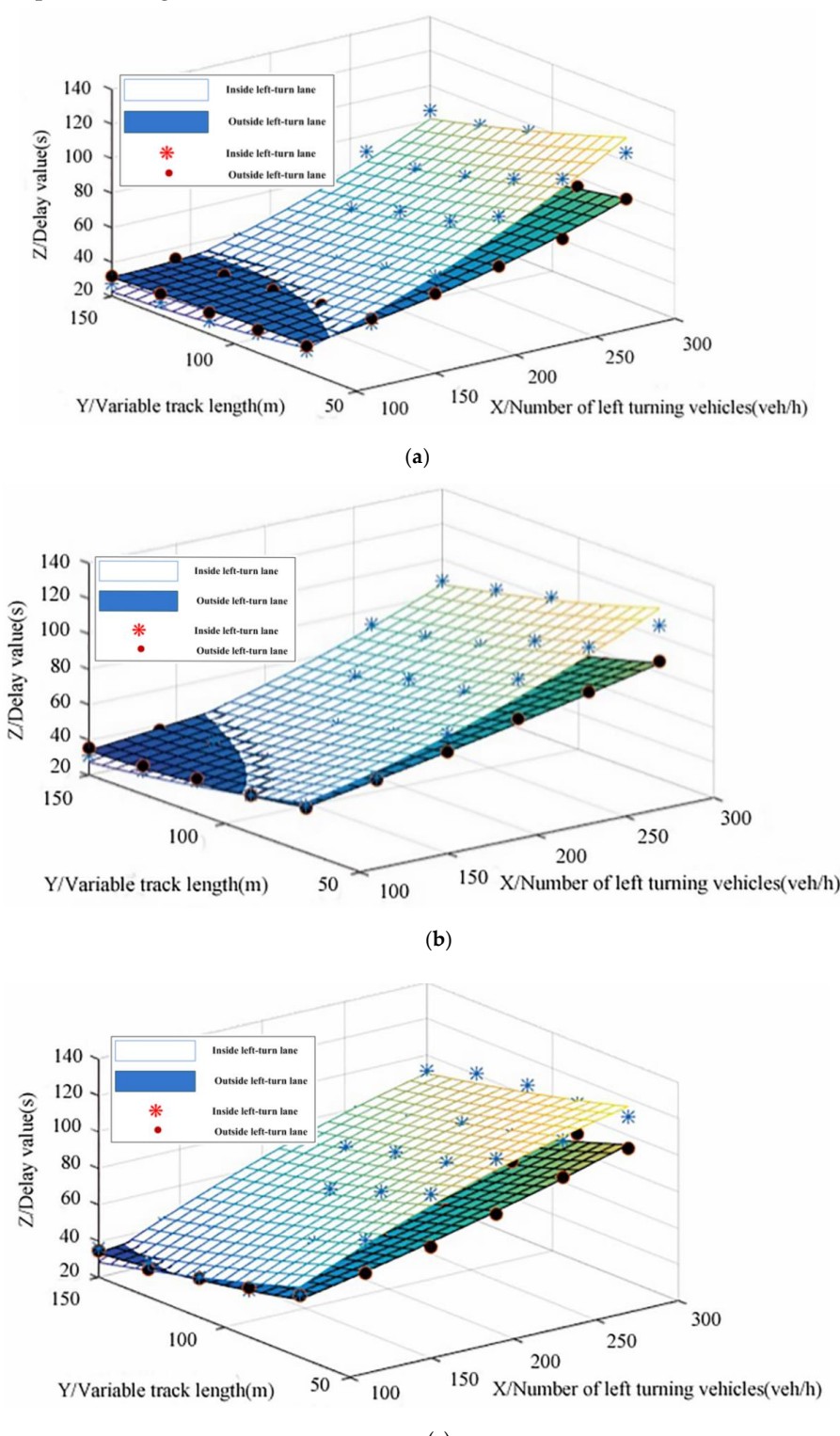

(a)

(b)

(c)

**Figure 2.** *Cont.*

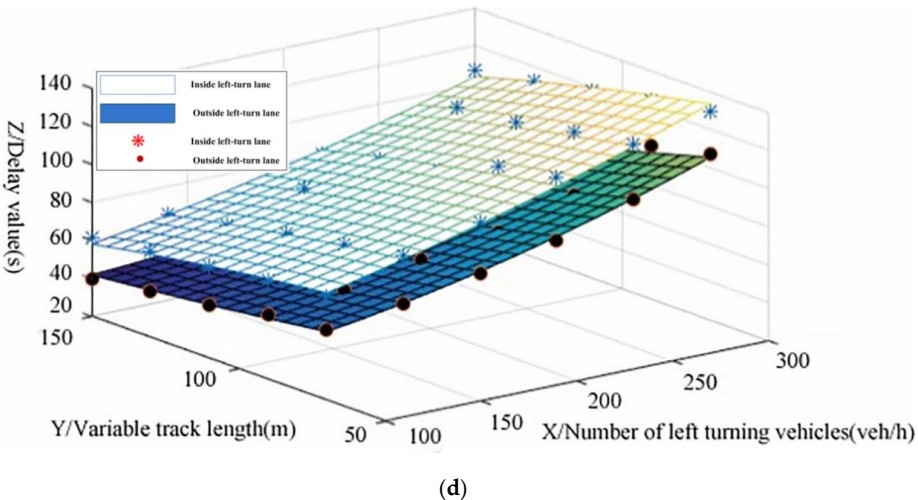

(**d**)

**Figure 2.** The influence of different numbers of left-turn vehicles and lengths of the weaving section on the delay. (**a**) The number of straight-going vehicles is 600 veh/h. (**b**) The number of straight-going vehicles is 750 veh/h. (**c**) The number of straight-going vehicles is 900 veh/h. (**d**) The number of straight-going vehicles is 1050 veh/h.

**Table 2.** The critical equation under different numbers of straight-going vehicles.

| Number of Straight-Going Vehicles | Curve Projection Equation |
| --- | --- |
| 600 | $y = (1.005e - 5)x^4 - 0.0045x^3 + 0.768x^2 - 56.62x + 1597$ |
| 750 | $y = 0.00061x^3 - 0.2182x^2 + 26.54x - 985.3$ |
| 900 | $y = 1.685x - 50.4$ |

It can be seen from Table 2 that the determination method based on the line of intersection is only applicable to the case where the number of straight-going vehicles is exactly 600, 750, or 900 veh/h. For more general cases, a delay model needs to be constructed for analysis.

### 3.2. BP Neural Network Delay Model

The BP neural network algorithm is an effective learning method for a multi-layer neural network. Its main feature is that signal is transmitted forward, and error is propagated backward. The error between the final output and the expected output of the network is minimized to achieve training purposes by continuously adjusting the network weight and threshold value.

#### 3.2.1. Data Normalization

In order to map all sample data to the same scale, it needs to be subjected to the maximum value normalization process, i.e., all sample data are mapped to a range of 0–1, as shown in Formula (1).

$$x_i = \frac{x'_i - x'_{\min}}{x'_{\max} - x'_{\min}} \tag{1}$$

where $x_i$ represents the normalized sample data, that is, the input data of the neural network model; $x'_i$ represents the original sample data; $x'_{\min}$ represents the minimum value in the original sample data; $x'_{\max}$ represents the maximum value in the original sample data.

#### 3.2.2. Network Structure Construction

In this study, a three-layer BP neural network with a single hidden layer is selected, as shown in Figure 3.

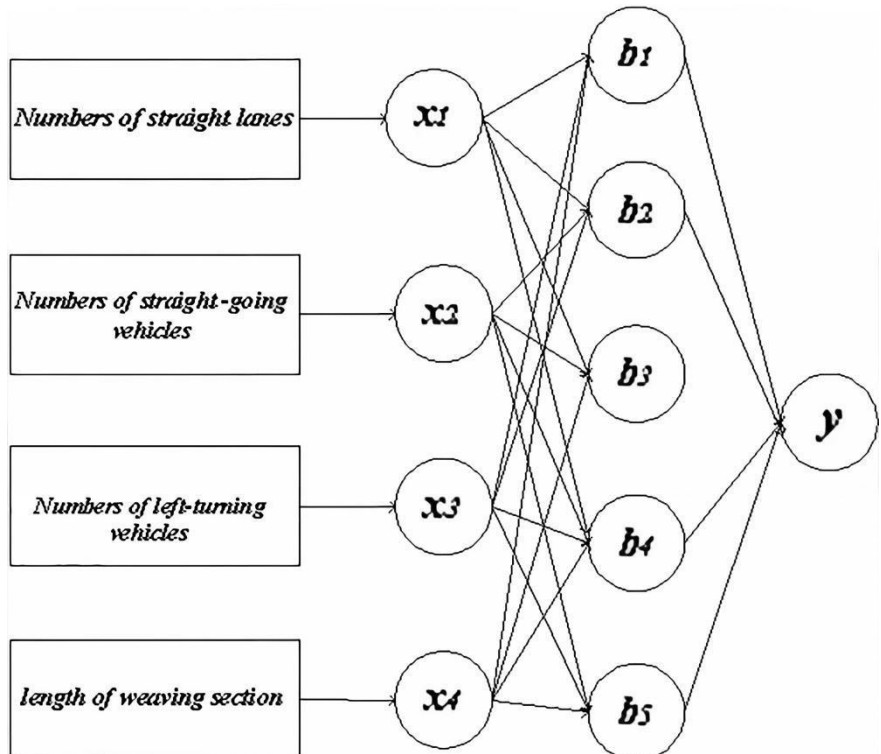

**Figure 3.** BP neural network structure diagram.

The number of neurons in the input layer is the number of input sample data types. Additionally, the number of neurons in the input layer is four because there are four influential factors. The output layer outputs one prediction result, so the number of neurons is one. The number of neurons in the hidden layer is calculated according to Formula (2):

$$j = \left[\sqrt{i + k} + a\right] \tag{2}$$

where $j$ represents the number of neurons in the hidden layer; $i$ represents the number of neurons in the input layer; $k$ represents the number of neurons in the output layer; $a$ represents a constant between 0–10, where three is taken according to the complexity of the problem (Hu Y, Fan Y Y, and Zhang X W, 2019). Thus, the number of hidden layer neurons is five.

### 3.2.3. Network Training

Since 240 pairs of delayed data are obtained in inside and outside simulation experiments, the number of training sample pairs of the model is 240. The vector $X$ is composed of four normalized influential factor data and used as input samples, as shown in Formula (3). The delay value y obtained by the simulation is taken as the output layer. When mapping from the input layer to hidden layer, the input value of each neuron in the hidden layer is calculated according to Formula (4).

$$X = [x_1, x_2, x_3, x_4] \tag{3}$$

$$b'_j = \sum_{i=1}^{4} w_{ij} x_i - \theta_j \tag{4}$$

In Formula (4), $w_{ij}$ represents the connection weight value from the input layer to the hidden layer; $\theta_j$ represents the hidden layer's threshold value.

Because of the nonlinear relationship between input sample influential factors and output sample delay, the activation function should be introduced to overcome the linear

model's limitations. Sigmoid is selected as the activation function, as shown in Formula (5). The output value $b_j$ of the hidden layer can be obtained by substituting Formula (4) into the activation function, as shown in Formula (6). When the hidden layer is mapped to the output layer, the input value of the output layer is calculated according to Formula (7).

$$f(x) = \frac{1}{1 + e^{-x}} \tag{5}$$

$$b_j = \frac{1}{1 + \exp\left(-\sum\limits_{i=1}^{4} w_{ij}x_i + \theta_j\right)} \tag{6}$$

$$y' = \sum_{j=1}^{5} v_j b_j - \alpha \tag{7}$$

In the formula, $v_j$ represents the connection weight value from the hidden layer to the output layer; $\alpha$ represents the threshold value of the output layer. Similarly, the output value of output layer is calculated according to Formula (8).

$$y'' = \frac{1}{1 + \exp\left(-\sum\limits_{j=1}^{5} v_j b_j + \alpha\right)} \tag{8}$$

The error of a pair of training samples is shown in Formula (9) in this case.

$$E = \frac{1}{2}(y - y'')^2 \tag{9}$$

The gradient reduction method is used to adjust the weight value and threshold value of each layer so that the error decreases along the gradient. The adjustment weight value from the hidden layer to the output layer and the threshold value of the output layer is shown in Formulas (10) and (11), respectively.

$$\triangle v_j = -\eta \frac{\partial E}{\partial v_j} = -\eta g b_j \tag{10}$$

$$\triangle \alpha = -\eta g \tag{11}$$

where $\eta$ represents the training rate, and its value determines how far the weight value can move in the direction of the error drop gradient in a small range. $\eta$ generally ranges from 0.01 to 0.8, and here, it is 0.02. $g$ is calculated according to Formula (12) and is the correction error of the layer's output.

$$g = (y - y'') \cdot y'' \cdot (1 - y'') \tag{12}$$

In the same way, the adjustment weight value from the input layer to the hidden layer and the threshold value of the hidden layer can be obtained, as shown in Formulas (13) and (14). The correction error $e_j$ of the hidden layer is calculated according to Formula (15).

$$\triangle w_{ij} = -\eta e_j x_i \tag{13}$$

$$\triangle \theta_j = -\eta e_j \tag{14}$$

$$e_j = b_j(1 - b_j)\sum_{i=1}^{4} v_j g \tag{15}$$

According to the adjustment value, the corresponding weight value and the threshold value of the next iteration are adjusted to complete an iteration process. In this study, two

sets of BP neural network models are established for two situations of inside and outside left-turn lanes. The set number of training is 1000.

## 4. Simulation Experiment

### 4.1. Influential Factor

For the traditional inside left-turn lane scheme, the following three basic characteristics are easily found by the field observation and preliminary simulation analysis: (1) When many left-turn vehicles are in the upstream outside lane, they need to change lanes inward to enter the target turning lane. This process will result in interweaving between left-turn vehicles and straight-going vehicles, and the frequency of interweaving also increases with the increase in straight-going vehicles. (2) When the number of straight-going lanes is large, the left-turn vehicles in the outside lane need to change lanes several times before entering the target lane, which also increases the chance of interweaving. (3) When the length of the weaving section is short, the left-turn vehicles in the outside lane need to make multiple lane changes within a short distance, which also greatly increases the probability of traffic conflicts. In view of the above situation, if the left-turn lane is placed outside, the interweaving and conflict of vehicles can be reduced to a certain extent, which will reduce traffic delay at the entrance, as shown in Figure 4. Therefore, the four indicators are the number of straight-going lanes, the number of straight-going vehicles in the entrance, the number of left-turn vehicles in the outside lane, and the length of the weaving section. Additionally, they are selected as the main influential factors for the inside or outside left-turn lanes.

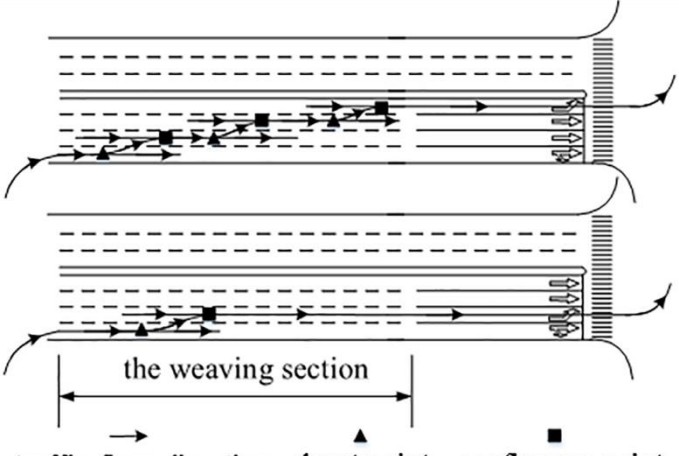

**Figure 4.** The weaving of traffic flow in inside and outside left-turn lanes.

### 4.2. Experimental Scheme

Using VISSIM, which can automatically identify the base map of actual road intersection, we can simulate the traffic situation in the left-turn lane as soon as we put in the relevant data about the length of the intersection, the number of lanes, the width of the lane, the layout of the sidewalk, and the setting of the parking line and traffic signs.

The f experimental scheme consists of taking the intersection of Xinlong St. and Shiji Road in Shenyang as an example. This simulation experiment optimizes the running direction of the intersection. It will adjust the direction of Lane 1 and Lane 2 of the south entrance to reduce the conflict of the actual intersection, which can achieve the purpose of this simulation optimization. The experiment is shown in Figure 5:

a.   Import the base map into VISSIM and set the correct scale;
b.   Set the road sections according to the direction of actual traffic, and then connect the entry and exit routes by connectors in accordance with the way that vehicles pass in real life;

c.  Input the survey data into the system of VISSIM traffic to calibrate parameters, including traffic volume, traffic composition, path decision setting, signal timing scheme, and traffic signal lights;
d.  Set the data detector and then select result output items, such as delay time, journey time, and traffic data in the system of VISSIM.

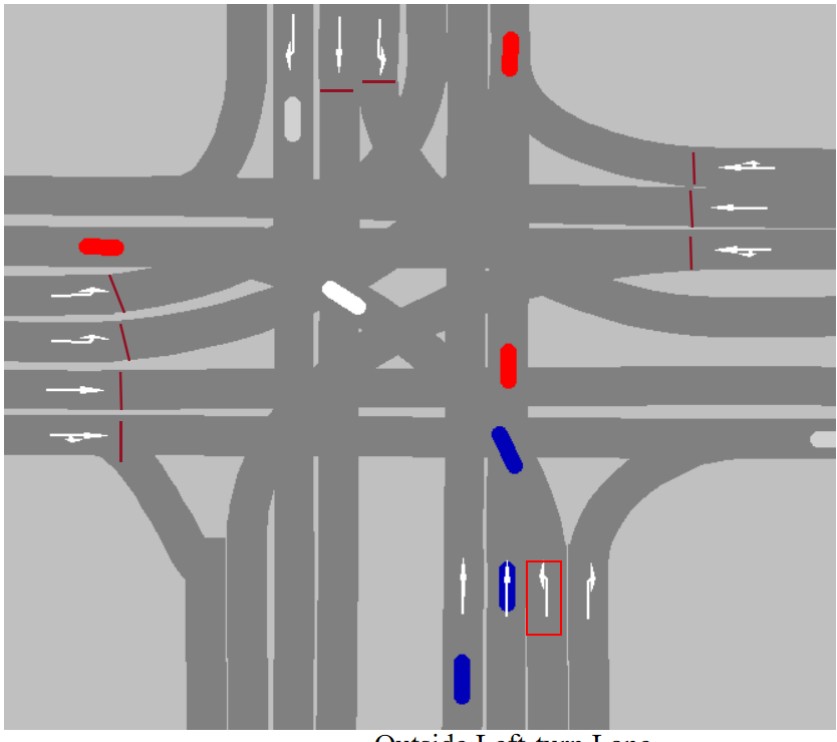

Outside Left-turn Lane

**Figure 5.** Simulation process.

After completing the above four steps, a set of data will be gained. Then, those data are input into the VISSIM system. Accordingly, we can conduct experiments by simulating the scene of the signal-controlled intersection many times. In this research, three simulation experiments were carried out by selecting different sets of data at random, and the average data of the results of the three experiments were adopted as the final result. The running time and simulation time are fixed in the three experiments. In each experiment, the VISSSIM system ran for at least 300 s but no more than 3900 s, and then the system was set to simulate the scene of interactions for one hour. After making the first model, the volume of traffic was adjusted at each entrance. The adjustment method is to keep the proportion of traffic volume in different road sections unchanged and reduce the number of vehicles. Theoretically, the simulation experiment cannot end until the average delay value decreases as expected. Because of the limited time, only three experiments were conducted in this research. The experimental procedure is shown in Figure 6.

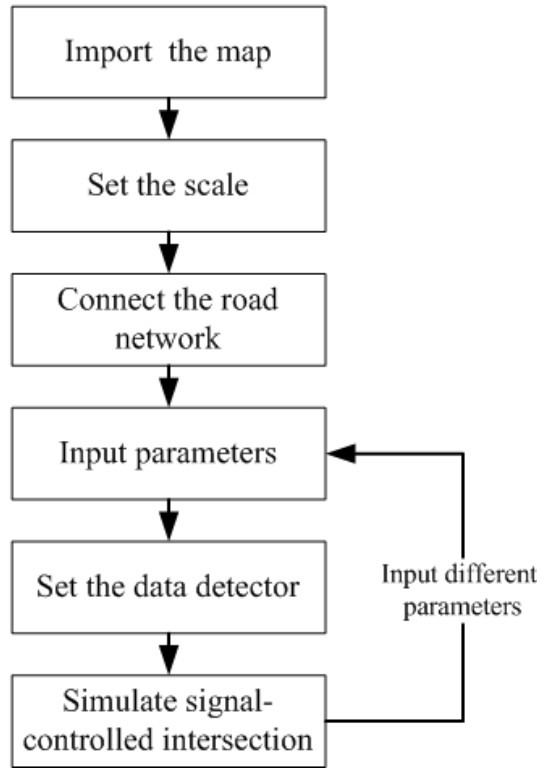

**Figure 6.** Experimental procedure.

*4.3. Parameter Calibration*

Based on the measured traffic data, some parameters (the average parking distance, the number of observable vehicles, and so on) of the car-following model, lane change model, and route decision model in VISSIM are calibrated and adjusted to make the simulated traffic volume close to the actual measurement (Yao R. H. and Wang D. H. 2009). In the subsequent simulation experiments, the calibrated model parameters are used for the simulation analysis.

To the calibrated model can simulate the basic operation status of road traffic and simulate the actual road conditions, such as signal timing scheme and geometric elements of the road, including the number of lanes at the entrance, the width of each lane, and the length of the weaving section. The simulation system provides a perfect car-following model and lane-changing model. Users can change parameters in the model parameters according to the actual situation so that the state of the vehicle is closer to the actual situation. The following will describe the specifics of the experiments in terms of route decision, expected speed, and vehicle composition:

a.      Routing Decision

In VISSIM, the traffic flow is generated by the edge of the road network and is collected by relevant upstream sections. The total traffic volumes are defined for all links and time intervals in vehicles, and then the routing decision is made for each lane. In VISSIM, "routing decision" is used to set the relative traffic flow ratio in each direction of the entrance. The traffic volume of roads with numerous vehicles merging and diverging is allocated according to the actual proportion of vehicles turning left, going straight, and turning right. In addition, the traffic flow in the mixed lane is distributed in different directions according to the actual proportion of vehicles turning left turn, going through, or turning right turn.

b.      Traffic Compositions

The traffic composition is used to define the vehicle mix of each input flow. It can help us determine the types of vehicles, relative flow of each car, and desired speed of each car. As for the type of vehicles, cranes and trolleys are chosen at each entrance lane of the intersection here, and the number of them is calibrated by the measured data.

c.  Desired Speed

The velocity data of different samples measured in the field have little difference. According to the data from samples, the desired speed of trolleys and cranes in the VISSIM system are set between 40 km/h and 60 km/h and between 20 km/h and 25 km/h, respectively.

*4.4. Experiment Conditions*

Due to the limited range of traffic data, which cannot completely cover all combinations of different setting locations of the left-turn lane, different lengths of the weaving section, different numbers of straight-going lanes, and different traffic volumes, traffic simulation data are used for research. The simulation data are based on the data of the intersection of Xinlong St. and Shiji Rd. and the intersection of Xinggong St. and Shenliao Rd. in Shenyang during the two peak periods of 7:00–8:00 and 17:00–18:00 on 5–6 July 2019. The simulation data are based on the expanded hypothetical data. Under the simulation model of different numbers of straight-going lanes, lengths of weaving section, and locations of the left-turn lane, respectively, the traffic delay data of the entrance lane are obtained by changing the number of straight-going and left-turn vehicles under different combinations of influential factors. The road traffic parameters of the simulation experiment are shown in Table 3. Each simulation time was 1 h (300~3900 s). Considering the influence of random factors, the average delay is obtained after three simulation experiments for the same road traffic conditions in each group. A total of 1440 simulation experiments are conducted in this research.

**Table 3.** The road traffic parameters of simulation experiment.

| Road Traffic Conditions | Lower Limit | Upper Limit | Steps | Number of Groups |
|---|---|---|---|---|
| Number of straight-going lanes | 2 | 3 | 1 | 2 |
| Length of weaving section/m | 70 | 150 | 20 | 5 |
| Number of left-turn vehicles/(veh/h) | 100 | 300 | 40 | 6 |
| Number of straight-going vehicles/(veh/h) | 600 | 1050 | 150 | 4 |
| Location of left-turn lane | inside | outside | - | 2 |

## 5. Results

*5.1. BP Neural Network Prediction Results*

Neural network delay analysis and model verification are performed based on peak-hour traffic parameters which are shown in Table 4.

**Table 4.** Traffic parameters of the entrance.

| Entrance Road Number | Number of Left-Turn Vehicles (veh/h) | | Number of Straight-Going Vehicles (veh/h) | | Location of Left-Turn Lane | Length of Weaving Section (m) | Number of Straight-Going Lanes |
|---|---|---|---|---|---|---|---|
| | Car | Large Vehicles | Car | Large Vehicles | | | |
| 1 | 480 | 48 | 609 | 123 | inside | 160 | 2 |
| 2 | 235 | 24 | 1809 | 131 | inside | 93 | 2 |
| 3 | 153 | 19 | 845 | 13 | outside | 90 | 2 |
| 4 | 391 | 41 | 1346 | 97 | outside | 100 | 3 |

The data of left-turn vehicles, straight-going vehicles, number of straight-going lanes, and length of the weaving section of four groups of intersections are normalized and input into the BP neural network model. The delay model value is calculated and obtained through the program. The left-turn lane at entrances 1 and 2 is changed to outside, and entrance lanes 3 and 4 are changed from outside to inside, while other parameters remain

unchanged. Then, the delay model value after changing the location of the left-turn lane can be obtained when parameters are input into the model, as shown in Table 5. The relationship between the mean square error and the number of training times corresponding to the inside schemes of the four left-turn lanes at entrance roads is shown in Figure 7. The law of change of the mean square error of the outside scheme is similar to it.

**Table 5.** The model and simulation values of delay in inside and outside left-turn lanes.

| Entrance Road Number | Inside Delay(s) | | Outside Delay(s) | |
|---|---|---|---|---|
| | Model Value | Simulation Value | Model Value | Simulation Value |
| 1 | 40.6 | 39.4 | 48.7 | 50.7 |
| 2 | 60.3 | 58.7 | 73.6 | 71.5 |
| 3 | 86.6 | 88.7 | 59.0 | 59.6 |
| 4 | 98.5 | 96.5 | 70.9 | 71.6 |

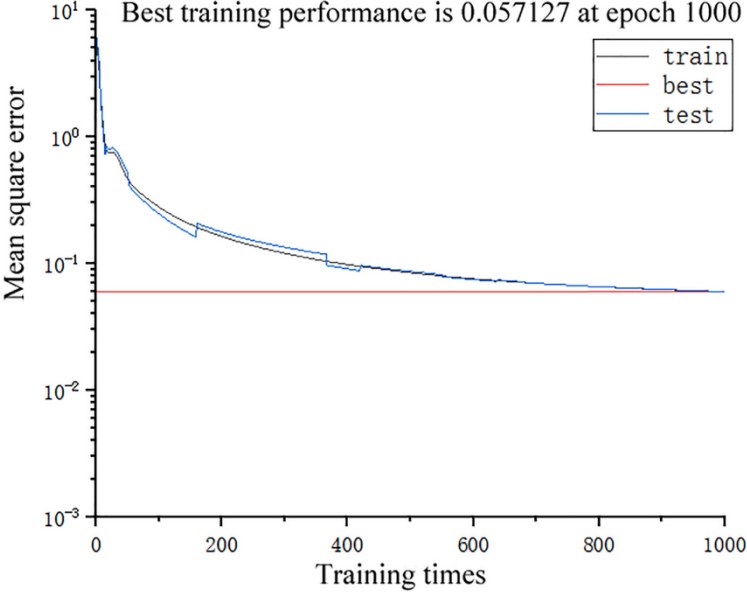

**Figure 7.** Relationship between mean square error and training times.

*5.2. Simulation Verification with VISSIM*

In order to verify the reliability of the BP neural network model for the delay analysis, the VISSIM simulation model is constructed for the above four intersection entrance lanes. The actual traffic data are input into the simulation model under the conditions of inside and outside left-turn lanes, respectively. The delay results obtained through the simulation analysis are shown in the simulation values in Table 5, and the delay comparison is shown in Figure 8.

It can be seen from Figure 5 that the delay in the inside left-turn lanes of entrances 1 and 2 is lower than the outside delay, and the delay of the outside left-turn lanes of entrances 3 and 4 is lower than the inside delay in the BP neural network model and VISSIM simulation model. The results show that entrance roads 1 and 2 are suitable for the conventional inside left-turn lane scheme, while entrance roads 3 and 4 are more suitable for the outside left-turn lane scheme. Additionally, the result is consistent with the realistic scheme. Moreover, Figure 6 also shows that the calculated values of the BP neural network model of each group of entrance delays are in good agreement with VISSIM simulation values.

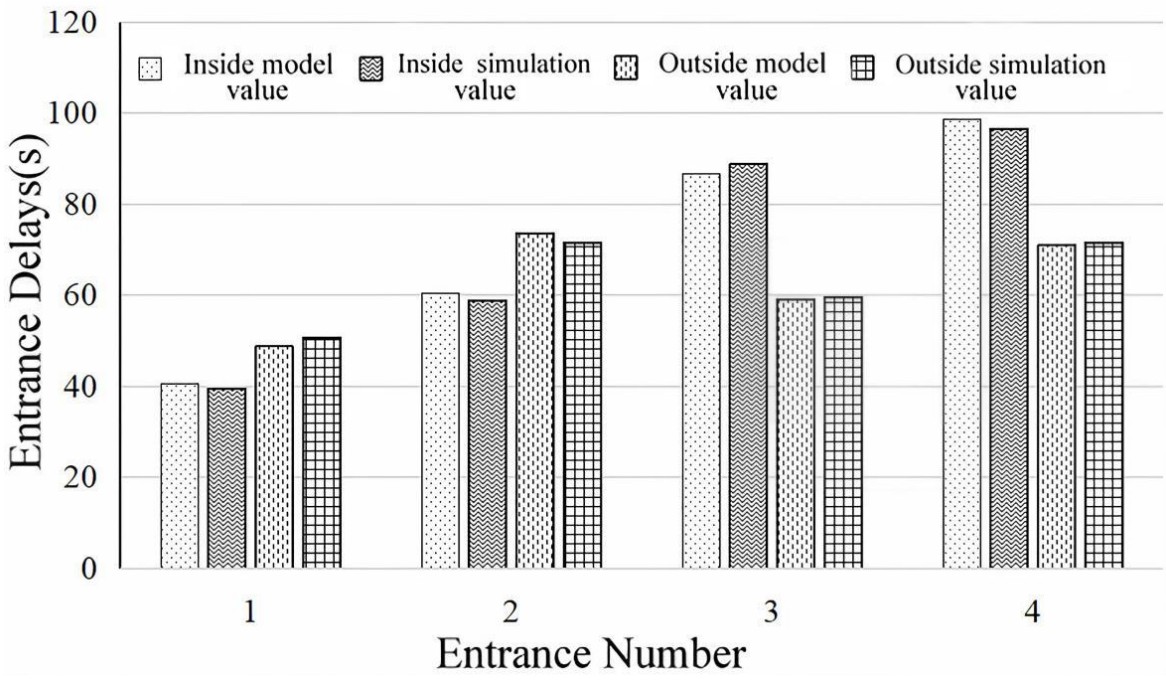

**Figure 8.** Delay contrast of model value and simulation value.

## 6. Discussion

Many scholars have conducted a lot of research on the settings of left-turn lanes, but few on the outside of left-turn lanes. Earlier scholars mostly focused on the saturated flow rate of multiple left-turn lanes Ronghan Y. (2020) [18], operational efficiency, safety, etc. (Sun H. et al., 2013) [19] Recently, academics have begun to carry out studies on the impact of outside left-turn lanes on the operation efficiency and safety of intersections (Junjie C. (2021) [20]). However, they seldom mention the determination method of the outside left-turn lane and the lack of quantitative research on the location of left-turn lanes under the influence of multiple factors. Although optimizing left-turn lanes through signal timing has proven to be effective in improving the capacity of left-turn lanes, improving the traffic capacity of a left-turn lane at an intersection cannot fully meet the demand of vehicles passing the left-turn lane. Additionally, the longer time it takes to pass the road caused by a left turn has attracted more and more attention. In this paper, the number of straight-going lanes, the number of straight-going vehicles in the entrance, the number of left-turn vehicles in the outside lane, and the length of the weaving section are integrated. The BP neural network delay model is established to analyze the delay under different conditions, and the judgment method of the outside left-turn lane is obtained.

With an in-depth study of the left-turn lane, it is not difficult to find that its location is particularly important for traffic (Hu, Z.A. et al. (2018)) [21]. In traditional research, the position of the left-turn lane is usually determined by the change of a single factor. In this paper, multiple factors influencing the position of the left-turn lane abroad are qualitatively analyzed. The conditions of traffic in the left-turn lane abroad and other lanes were compared, and the consequences illustrated the effectiveness of improving left-turn lanes for improvements in transportation.

To solve the problem of setting left-turn lanes, scholars have proposed the combination of VISSIM and grayscale, signal control strategies, and other algorithms (Dmitry Likhachev et al., 2019) [22]. As for the problem, other scholars together put forward a method that the combination of VISSIM and a BP neural network is used to determine the position of the left-turn lane in 2020. In this paper, a BP neural network is used to predict the data when there is a traffic jam in the left-turn lane. The predicted data are compared with the simulated data obtained by VISSIM, which fully shows the feasibility of BP neural network prediction.

## 7. Conclusions

(1) From the perspective of delay, the outside left-turn lane is suitable for the following situation: more left-turn vehicles in the outside lane, more straight-going vehicles on the section, a shorter length of upstream weaving section, and more straight-going lanes at the intersection entrance.

(2) According to the number of lanes, straight-going vehicles, left-turn vehicles, and the length of the weaving section, the delay model of the BP neural network can be used to calculate the traffic delay corresponding to the scheme of inside or outside left-turn lanes, respectively, and the analysis results of the model are in good agreement with the simulation results. The appropriate location of the left-turn lane can be determined through delay comparison.

Because this study involves the delay analysis under the comprehensive influence of multiple factors, it is difficult to find intersection samples that meet the full combination of all influential factors in reality, and the location of conventional left-turn lanes is exclusive. Thus, this study is based on simulation experimental data rather than measured data. However, the general principles and methods adopted and the regular characteristics obtained from the analysis can still be used for reference in similar research. The reliability of the simulation experiments needs to be verified by more large-scale cases.

**Author Contributions:** Conceptualization, Y.C., D.J. and X.L.; methodology, Y.C., D.J. and X.L.; software, X.L.; investigation, Y.C.; validation, Y.C., D.J. and X.L.; visualization, Y.C. and D.J.; writing—original draft preparation, Y.C., D.J. and X.L. All authors have read and agreed to the published version of the manuscript.

**Funding:** This research was supported by the Scientific Research Funding Project of the Liaoning Provincial Education Department in 2020 (funder: Department of Education of Liaoning Province, grant No. JDL2020017), the Educational Science Planning Project of Liaoning Province (funder: Liaoning Provincial Education Science Planning Leading Group, grant No. JG20DB69), the Research Project on Economic and Social Development of Liaoning Province in 2022 by the Liaoning Provincial Federation Social Science Circles (funder: Liaoning Provincial Federation Social Science Circles, grant No. 2022lslybkt-022), 2021 Project of the Dalian Academy of Social Sciences (funder: Dalian Academy of Social Sciences, grant No. 2021dlsky050), the Education Quality Improvement Project for Post-graduate of Dalian Jiaotong University (funder: Dalian Jiaotong University), and the Teaching Reform Research Project for Undergraduate of Dalian Jiaotong University (funder: Dalian Jiaotong University).

**Institutional Review Board Statement:** Not applicable.

**Informed Consent Statement:** Not applicable.

**Data Availability Statement:** Not applicable.

**Acknowledgments:** We appreciate the nine postgraduate students at the corresponding author's university for investigating and collecting the traffic data. We are also grateful to the editors and anonymous reviewers for their suggestions and comments.

**Conflicts of Interest:** The authors declare no conflict of interest.

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
