# Peer review of "Influential Factors and Determination Method of Unconventional Outside Left-Turn Lanes Based on a BP Neural Network"

_applsci, doi:10.3390/app12126026_

Round 1

Reviewer 1 Report

This research was focused on investigating the influential factors and the determination method of unconventional outside left-turn lane to reduce conflicts between left-turn and straight-going vehicles at the upstream of intersection entrance. Research results revealed that when the number of left-turn vehicles in on the outside lane and straight-going vehicles at the entrance or the straight-going lanes are larger while the length of the waving section is shorter, the daily reduction effect of unconventional outside left-turn lane will be more obvious. The reviewer believes that the current version of the manuscript is not yet ready for publication; the authors are encouraged to consider the following comments and suggestion and revise the manuscript accordingly.

1. The authors should improve the Abstract section. Currently it is clear and very difficult to provide a summary of the study. The Abstract section should be focused on presenting what the problem is, what the solution is, and what the conclusion is.

2. The authors need to use the journal’s template to prepare their manuscript. Additionally, the authors should have a professional editor go through the manuscript to remove grammatical and spelling errors. Currently, there are many grammar errors and it is very difficult to read.

3. The authors should consider splitting the Introduction section into two sections, including an Introduction section and a Background section. The introduction section should focus on introducing the research objectives and research questions to be answered, while the Background section should focus on literature review of related works and presenting the research gap. The authors need to review more literature.

4. The authors mentioned that some of the parameters of the car-following mode, lane change model, and route decision model in VISSIM are calibrated and adjusted to make the simulated traffic volume close to actual measurement. What is the calibration and adjustment process? How is the calibration and adjustment being completed? The authors need to provide more information on this. To the reviewer, this is the one of the most critical research field and the authors have failed to provide the information.

5. The authors need to discuss more about the case study. The authors need to provide more insightful discussion based on the interpretation of the case study.

6. The authors need to go through the algorithms to ensure all symbols are appropriately denoted.

7. All figures need to be improved. The reviewer has to zoom in at least 200% to be able to read them. If at all possible, please use vector images.

Reviewer 2 Report

In the paper entitled “Influential Factors and Determination Method of Unconventional Outside Left-turn Lane Based on BP Neural Network” the authors present a queantitative research on the influential factors and setting conditions of the outside left-turn lane. However, as it stands, the article cannot be published in this journal unless a number of changes and improvements are made.

1) In general, the organisation and structure of the article should be improved, as it is sometimes very ambiguous.

2) The introduction does not specify the objectives of the work presented, and the structure of the article is not specified.

3) It would be necessary to add an initial scheme of how the experiment has been carried out.

 4) In addition, the description of the method should be improved by better explaining the parameters used.

5) the mathematical context is weakly explained and the novelty and improvement of the method worked on in the article is not seen.

8) Could the authors add more descriptive graphical results for Table 3 and Table 4.

9) Finally, the authors should add more papers in the introduction and bibliography describing recent and relevant related research works.

10) The conclusions are inconclusive and the contribution of the article remains ambiguous.

11)There is a point 7.Conclusions repeated and without content before the references.

Round 2

Reviewer 1 Report

The authors have address all my comments.

Author Response

Dear expert,

Thank you for your suggestions.

Reviewer 2 Report

THE ARTICLE HAS IMPROVED CONSIDERABLY, BUT I STILL CONSIDER THAT THE FOLLOWING IMPROVEMENTS NEED TO BE MADE:

- THE STRUCTURE IS STILL DEFICIENT. THERE SHOULD BE AN INTRODUCTION, CASE STUDY, METHODOLOGY, EXPERIMENT, RESULTS, DISCUSSION OF RESULTS AND CONCLUSIONS (IN THAT ORDER).

- WHEN I SAY THAT AN EXPERIMENTAL SCHEME SHOULD BE ADDED, I AM ALSO REFERRING TO A DRAWING OR GRAPHIC SCHEME (NOT JUST TEXT) AS THIS WOULD FACILITATE THE UNDERSTANDING OF THE WORK.

-FINALLY, ENGLISH SHOULD BE REVIEWED BY AN ENGLISH LANGUAGE EXPERT.

Author Response

(The authors gave the same response as above.)

Round 3

Reviewer 2 Report

The manuscript has been sufficiently improved to warrant publication in Applied Sciences.